# Conserved mammalian modularity of quantitative trait loci revealed human functional orthologs in blood pressure control

**Alan Y. Deng**\*, **Annie Ménard**

Department of Medicine, Research Centre, CRCHUM (Centre hospitalier de l'Université de Montréal), Université de Montréal, Montréal, Québec, Canada

\* alan.deng@umontreal.ca

## Abstract

Genome-wide association studies (GWAS) have routinely detected human quantitative trait loci (QTLs) for complex traits. Viewing that most GWAS single nucleotide polymorphisms (SNPs) are found in non-coding regions unrelated to the physiology of a polygenic trait of interest, a vital question to answer is whether or not any of these SNPs can functionally alter the phenotype with which it is associated. The study of blood pressure (BP) is a case in point. Conserved mechanisms in controlling BP by modularity is now unifying differing mammalian orders in that understanding mechanisms in rodents is tantamount to revealing the same in humans, while overcoming experimental limitations imposed by human studies. As a proof of principle, we used BP QTLs from Dahl salt-sensitive rats (DSS) as substitutes to capture distinct human functional orthologs. 3 DSS BP QTLs are located into distinct genome regions and correspond to several human GWAS genes. Each of the QTLs independently exerted a major impact on BP *in vivo*. BP was functionally changed by normotensive alleles from each of these QTLs, and yet, the human GWAS SNPs do not exist in the rat. They cannot be responsible for physiological alterations in BP caused by these QTLs. These SNPs are genome emblems for QTLs nearby, rather than being QTLs *per se*, since they only emerged during primate evolution after BP-regulating mechanisms have been established. We then identified specific mutated coding domains that are conserved between rodents and humans and that may implicate different steps of a common pathway or separate pathways.

## Introduction

The formation of a 4-chambered heart in mammals [1] has co-opted mechanisms regulating blood pressure so that many living land mammals including rodents and humans reach a similar level [2]. This blood pressure conservation is sustained among differing orders of mammals, despite differences in other physiology characters (e.g. heights, reproductive patterns). The only way this can happen is that basic mechanisms regulating blood pressure (BP) must have been formulated in common ancestors of rodents and humans before 90 million years

**Funding:** This study is funded by Heart and Stroke Foundation of Canada, number 12345. AYD

received this award. The funder had no role in study design, data collection and analysis, decision to publish, or preparation of the manuscript.

**Competing interests:** The authors have declared that no competing interests exist.

ago (www.timetree.org). That was before the human genus existed and before rodents diverged from primate ancestors [3, 4]. Modern humans surfaced only about 300,000 years ago [5], and not surprisingly, acquired same mechanisms in controlling BP from primate ancestors in parallel with rodents in an evolutionary tree of mammals [3, 4].

Our practical goal of conducting studies in any organism is to help us understand mechanisms of human hypertension pathogenesis [6]. Unknowing which mechanisms are involved in hypertension pathogeneses in general populations, genetic analysis is believed to be unrestrained by known BP physiologies and thus be a promising lead-in towards achieving it [7]. Thanks to genome-wide association studies (GWASs) [8], detecting human quantitative trait loci (QTLs) for BP became possible. Since then, an incremental gain on an ever-enlarging scale has statistically marked the vicinity of more than 900 BP QTLs by single nucleotide polymorphisms (SNPs) [9], even though a statistical association of a SNP is not a proof of its function on BP.

Total variance has been used in quantifying phenotypic variations in heterogeneous human populations. Since only a small fraction of total variance is due to QTLs' impact, and most is due to environmental effects [10], the actual magnitude of a physiological effect on BP from a single human QTL could not be known from GWAS. This difficulty is exacerbated by the fact that no human QTLs have been identified to be any component of a physiology system known to affect blood pressure [11, 12]. A physiological difference is obvious between locating a SNP marking a QTL nearby and identifying the QTL itself. Even identified, larger gaps exist between probable triggering pathways initiated by QTLs leading to BP control and known path-physiologies that execute it. Now we are no closer in understanding a pathogenesis for human polygenic hypertension than before the advent of GWASs [13]. Further realizing that most GWAS SNPs are located in non-coding regions with unknown functions, we focused on a pivotal question: 'can any of these GWAS SNPs affect blood pressure by function *in vivo*?

Due to experimental advantages using rodent models, mechanisms modulating blood pressure driven by function can be revealed, whereas human studies are limited to epidemiology. Because of conserved mechanisms between rodents and primates, studying BP regulating mechanisms in rodents is equivalent to revealing the same mechanisms in humans, while overcoming unavoidable limitations in human studies.

Recently, we used rodent QTLs as proxies to functionally capture distinct human QTLs [14]. 3 rodent BP QTLs resemble 3 specific human GWAS genes. Each of them independently showed a major impact on BP *in vivo*. BP was functionally lowered by normotensive alleles from each of these 3 QTLs, and yet, human GWAS non-coding SNPs do not exist in the rat. Thus, they cannot be involved in the physiological changes in BP caused by these QTLs that have been established before human and rodent ancestors diverged 90 million years ago. These non-coding SNPs are genome insignias earmarking human QTLs nearby, rather than being QTLs themselves. Together, 45 human GWAS genes may fall into 2 epistatic modules /2 common pathways of the BP homeostasis. Thus, QTL modularity explains redundancy of human BP QTLs in collectively controlling BP, and is conserved between rodents and humans.

Nevertheless, this initial work only covered a small section of the rat genome from 21 chromosomes. Here, we aimed at expanding the scope of rat QTL coverage to capture additional human GWAS genes in progressive stages. In the process, our new results have validated the reproducibility of our previous findings. We focused on distinct regions on DSS rat Chromosomes 7 and 8 that contain previously-unexplored rat blood pressure QTLs and human GWAS gene orthologs.

This paper reports results of additional studies in unifying the human and rodent physiology in BP control. (1) We extended functional captures of human QTLs to other genome

regions. (2) We grouped these human-relevant QTLs by function into modules. (3) We assessed the functional relevance of GWAS SNPs in BP control, in an *in vivo* model of their natural 'knock outs'. (4) We singled out functional candidates for several QTLs.

## Materials & methods

### Animals

Animal research has been approved by our institutional committee (CIPA, Comité institutionnel de protection des animaux) with approval number CM19037ADr. Protocols in animal experiments were approved by our institutional animal committee, CIPA. Our basic animal model is inbred hypertensive Dahl salt-sensitive (DSS) rats. Congenic strains were made by replacing distinct chromosome regions of DSS by those of inbred normotensive Lewis rats. Most congenic strains used in the current study are synthesized from our previous work [15]. Experimented animals were monitored daily. Euthanasia was done with isofurane at 4% and O2 at 1L/min. The chromosome segment congenically 'knocked in' defines the interval harboring each of multiple QTLs. The chromosome regions containing them are delimited by microsatellite markers.

### Experimental protocols and analyses

They are the same as documented previously [14, 15]. In brief, male rats were weaned at 21 days after birth, fed on a low-salt diet (0.2% NaCl), then a high-salt diet (2% NaCl) starting at 35 days of age until the experiment was done. Telemetry probes were implanted into the rats of 56 days old. Since systolic and diastolic pressures were consistent with mean arterial pressures (MAP) of all the strains [16, 17], only their MAPs are provided for simplicity. The power and sample size were calculated to be sufficient [18].

Repeated measures' analysis of variance (ANOVA) followed by the Dunnett's test was used to compare the significance in a difference or a lack of it in a blood pressure component between a congenic strain and the DSS parental strain. The Dunnett correction takes into account multiple comparisons as well as sample sizes among the comparing groups. In the analysis, an averaged BP component was compared for each day for the period of measurement among the strains.

Because blood pressures of rats were measured continuously for 2 weeks with one reading at every 2 minutes, the numbers given at the bottom of Fig 1 represent only averaged values of mean arterial pressures for a strain during 2 weeks. A Dunnett value including "<" given in each comparison between congenic and DSS strains was the most conservative P value among all the days of comparisons.

## Results

### (A) Functional proxy tool and conceptual definitions are as follows

(a). A congenic strain refers to a homogeneous strain in which a sole chromosome segment from the recipient strain has been replaced by that of the donor strain, while retaining the rest of the genome as that of the recipient. When BP changes as a result, a QTL functionally exists in the replaced segment. In essence, the congenic strategy is similar to that of 'knock-in' [7], and is thus termed congenic knock in genetics. In order to visualize physiological effects from BP QTLs, our congenic knock in genetics was done in the DSS genetic background that has lost its genome buffering capacity in tampering BP fluctuations [19] and hypertension suppression [18, 20].

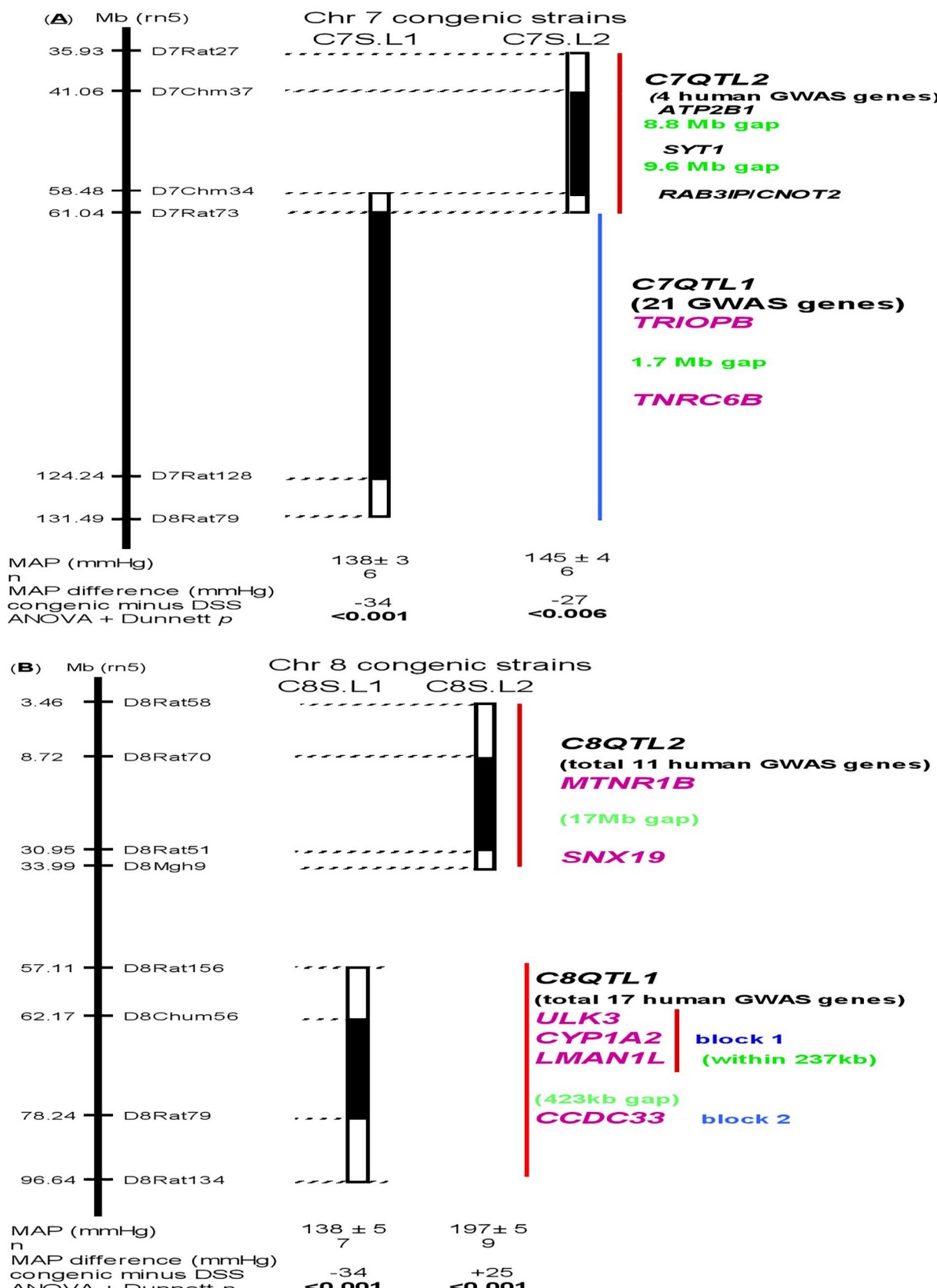

**Fig 1. Congenic knock in genetics defining BP QTLs *in vivo*.** A solid bar under congenic strains represents the Dahl salt-sensitive rat chromosome fragment that has been replaced by that of normotensive Lewis rats (S.L). Striped bars on ends of the solid bars denote the ambiguity of crossover breakpoints between markers [15]. Full names for abbreviated genes of interest are listed in the footnote for Table 1.

(b). Although a QTL is genetically localized to a genome segment containing multiple genes, the molecular basis of 1-gene-for-1 QTL is standardized during QTL identification [7]. This has been affirmed by *C17QTL1* as *CHRM3* encoding muscarinic cholinergic receptor 3 (M3R) [14, 21]. No combination is required for *C17QTL1* with any other QTLs to change BP. A missense mutation in *Chrm3* of Dahl salt-sensitive rats (DSS) alters its signaling and was primed for its identity as *C17QTL1*. The functional dosage in M3R signaling, not the *Chrm3* gene dose, determines the degree of hypertension pathogenesis [22]. The intergenic GWAS SNP nearby [8] is only a marker for the human QTL, not the QTL itself [14]. Thus, a 'common' SNP with a set minor allele frequency for human GWAS merely labels a nearby biological QTL that has a 'rare' functional variant. Whether or not the SNP was replicated in other GWAS by statistics [9] is irrelevant to its functional impact on BP, because depleting it has no effect on BP [21, 22].

**Table 1. Identification of human QTLs/GWAS genes from their functional proxy.**

| Rat QTL name | Magnitude of BP effects | Rat Gene | Mutation detected Lew/DSS | Change in amino acid (AA) Lew/DSS | Human GWAS SNP | Rat GWAS SNP ortholog | Closest human GWAS gene | # probable human missense mutation |
|---|---|---|---|---|---|---|---|---|
| *C7QTL1* | -41% | *Nxph4* | A571C | T191P | | | No GWAS gene | 13 |
| | | *Rdh16 (Rdh2)* | A67G G311C T746C | M23V R104T M249T | | | No GWAS gene | 10 |
| | | *Tac3 (Tac2)* | C162T | P55S | | | No GWAS gene | 6 |
| | | ***Triobp*** | G1491T | Q497H | rs4820296 rs12628603 rs1129448 | Non-existent Non-existent Existent | ***TRIOBP*** | 9 |
| | | ***Tnrc6b*** | DSS CAG insertion at 3784 | DSS Q insertion (1262) | rs470113 | Non-existent | ***TNRC6B*** | 15 |
| *C8QTL1* | -41% | ***Ulk3*** | A770G | H257R | rs6495122 | Non-existent | ***ULK3*** | 7 |
| | | ***Cyp1a2*** | G410A T1207C | R137H C403R | rs1378942 | Non-existent | ***CYP1A2*** | 16 |
| | | *Lman1l* | A64G A809G C1164A | S22G Q270R H388Q | | | No GWAS gene | 17 |
| | | ***Ccdc33*** | C1108A C2095T | P370T R699C | rs351157 rs4887123 rs94899 | All Non-existent | ***CCDC33*** | 9 |
| *C8QTL2* | +30% | ***Mtnr1b*** | T100C[†] | S34P | rs10830963 | Non-existent | ***MTNR1B*** | 15 |
| | | ***Snx19*** | C430A[*] | L144M | rs1050081 rs2276098 rs948086 | All Non-existent | ***SNX19*** | 18 |

QTLs are given in Fig 1. – or + in magnitude of BP effects in physiology indicates the effect of normotensive alleles for that QTL in decreasing or increasing BP. Genes in bold are QTL candidates for both rats and humans. All missense mutations have been confirmed by independent clonings of the genome segments containing them followed by sequencings.

[*] indicates confirmed missense mutations detected in our database and by the public rat genome database.

[†] The mutation showed up in our database only, but has not been confirmed by independent sequencing due to technical difficulties. DSS, Dahl salt-sensitive rat; Lew, Lewis rat. Number of human missense mutations indicates those with minor allele frequency of >0.04% from exon sequencings. ***Ccd33c,*** coiled-coil domain containing 33; ***Cyp1a2,*** cytochrome P450 family 1 subfamily A member 2; ***Lman1l,*** lectin, mannose binding 1 like; ***Mtnr1b,*** melatonin receptor 1B; ***Nxph4,*** neurexophilin 4; ***Rdh16,*** retinol dehydrogenase 16; ***Snx19,*** sorting nexin 19; ***Tac3,*** tachykinin precursor 3; ***Triobp,*** TRIO and F-actin binding protein; ***Tnrc6b,*** trinucleotide repeat containing adaptor 6B; ***Ulk3,*** unc-51 like kinase 3.

(c). Non-coding human GWAS SNPs [9] so far analyzed are by products of primate evolution, since they appeared only in primates not in rodents after they have evolutionarily diverged. This fact indicates that appearances of these SNPs are independent of the establishment of BP regulating mechanisms, which was already fixed before rodents and primates diverged (see introduction). Thus, such a non-coding GWAS SNP cannot be relevant to BP by function [14].

The above 3 considerations guide functional discovery of additional BP QTL candidates for humans in following sections. Among all DSS chromosome regions known to harbor BP QTLs [15], only those matching human QTL signals from GWAS [9] were analyzed here (S1 Table). We begin by functionally defining individual QTLs with human orthologs, followed by grouping them in epistatic modules. We then focused on molecular mechanisms of QTL actions and evaluated the functional impact of human non-coding GWAS SNPs in BP control by proxy.

## (B) QTL modularity on BP is functionally conserved between differing mammalian orders

The BP effect for one gene among hundreds detected in human GWAS [9] was calculated as a fraction of total BP variance and seemed miniscule [11]. However, the real physiological BP effect of a single QTL by function is considerably larger when viewed in homogeneity, by causality and alone [details were given in Table 1 in reference [10]].

*C7QTL* on DSS rat Chromosome 7 is a case in point [17], and is now re-designated as *C7QTL1* (Fig 1A). *C7QTL1* was not detected and statistically explained 0% of total variance in an $F_2$ population [23], yet was capable of functionally altering BP by 41% in the total difference between 2 parental rat strains *in vivo* [17]. The calculation is as follows. The congenic knock-in lowered blood pressure by 34 mmHg, or indicated by -34 (Fig 1A). The total BP difference between DSS and Lewis strains is 83 mmHg, i.e. 178 mmHg for DSS minus 95 mmHg for Lewis. Thus, the blood pressure effect of *C7QTL1* is calculated as -34/83 = -41% (i.e. BP-decreasing effect). By the same calculation, other 3 QTLs, *C7QTL2*, *C8QTL1*, and *C8QTL2*, singularly possess a major physiological BP effect of respective -33%, -41% and +30% (i.e. BP-increasing effect) (Fig 1, Table 1).

How do these 4 QTLs along with 24 others [21] functionally affect BP together *in vivo*? Summing them up cannot be physiological, since that would lower BP to below zero, whereas fractionating each QTL by total variance is not functional. Thus, a valid biological solution is needed on how they together can sustain BP *in vivo*. It turned out that epistatic modularity charts fundamental mechanisms of QTL actions on BP that are functional and physiological [10], i.e. their effects on BP are collectively additive only in modularity [15].

The *C8QTL1*-residing interval harbors 17 human GWAS genes [9] (Fig 1B) and the *C10QTL1*-residing interval [24] bears 3 human GWAS genes [9]. When the segment carrying *C8QTL1* was merged with that for *C10QTL1*, their combined BP did not exceed BP for either of them alone [15]. This epistasis means one QTL masking the effect of another on BP. Their coexistence in epistasis puts them in the same epistatic module (EM), i.e. EM1 [15]. Thus, QTL modularity on BP is conserved between the human GWAS genes and the rodent QTLs that proved their presence by function.

## (C) *C8QTL1* of DSS rats functionally corresponds to one of several human GWAS genes [9]

(**a**) Normotensive alleles of the rodent *C8QTL1* lower BP *in vivo* alone and independently of other QTLs (Fig 1B). The *C8QTL1*-harboring segment contains a total of 17 human GWAS

genes (S1 Table). Among them, 3, and 1 genes respectively can be grouped into 2 genome blocks, according to the existence of functional candidates from DSS rats harboring missense mutations (Fig 1B and Table 1). 2 intergenic GWAS SNPs, rs1378942 and rs6495122, define block 1, and several intronic SNPs mark block 2 (S1 Table). A distance of 423 kb separates block 1 from block 2 (Fig 1B) and implies 2 distinct QTLs in the region now defined to contain *C8QTL1*. To confirm this, further evidence of fine resolution by congenic knock in genetics [21] will be needed to trap each block and to isolate it from one another by function.

Regardless, both human GWAS and *in vivo* rodent studies converge to prove that one or several of GWAS signals statistically associated with BP may be capable of functionally altering BP. Conversely, *C8QTL1* has extended to humans beyond the confines of rodents, and functionally changed BP in a magnitude of 41% (Table 1) in the total difference between 2 contrasting parental strains. This is evident despite that how many human GWAS genes corresponding to how many rodent QTLs remain unresolved in the *C8QTL1*-residing region, which is still large (Fig 1B).

(**b**) Next, we examined the molecular basis that prioritize 5 functional candidates for the 2 blocks containing human GWAS genes (Fig 1B, Table 1). If one or some of these inter-and intronic GWAS SNPs would affect BP *in vivo*, they should exist in congenic strain C8S.L1 which served a functional proxy for human GWAS genes, since BP was lowered by normotensive *C8QTL1* alleles (Fig 1B). We first searched all these GWAS SNPs and no similar sequences were detected in the rat genome (S2 Table). No homologies were found for the mouse genome. The absence of these SNPs is not due to a possible anomaly in rat genome assembly. These SNPs and surrounding non-coding sequences are not conserved in rodents. The genome regions around the intergenic and intronic SNPs are not known to play a regulatory role in gene expressions.

Since the physiological change in BP occurred *in vivo* (Fig 1B), when these SNPs are naturally 'knocked out', the SNPs themselves cannot be responsible for altering BP by *C8QTL1*, and appears solely as a marker for the functional QTL nearby, not the QTL *per se*. This conclusion was further strengthened by the fact that these SNPs are by-products of primate evolution, rather than a demand in functionally controlling BP. These SNPs emerged only in certain simians and apes (S3 Table), long after primates and rodents have diverged, and long after BP-regulating mechanisms common between primates and rodents were already set and remained little changed since.

(**c**) We reasoned that physiologically, valid candidate genes have to be both conserved between the rat and humans as well as capable of potentially altering BP by function. The coding domains of five genes fulfils these 2 criteria. In block 1 (Fig 1B), *Ulk3*, *Cyp1a2*, and *Lman1l* are close to one another (S1 Table), and carry missense mutations, which were first detected in both our genome data base [25] and the public rat genome data base (RGD) [26] (Table 1). We then individually cloned and sequenced all the exons and intron-exon junctions of the 3 genes. The same missense mutations were then confirmed to alter their protein structures and may have a functional impact. Thus, their coding structures are strongest functional candidates for corresponding 3 human GWAS genes.

We next searched human coding domain data bases [27], and found 7, 16 and 7 missense mutations in human *ULK3*, *CYP1A2*, and *LMAN1L* respectively (S4 Table). Although the same missense mutations were not found in humans as in rats, various missense mutations throughout their protein structures might have functional implications.

(**d**) One functional candidate was found for block 2 in the *C8QTL1*-residing interval (Fig 1B), because *Ccdc33* contains missense mutations in both rats (Table 1) and humans (S4 Table). In contrast, the intronic GWAS SNPs in *CCDC33* (S1 Table) only exist in humans and other primates (S3 Table), but are 'knocked out' in rodents (S2 Table). Once again, these non-

coding GWAS SNPs can only be markers for the potential QTL nearby, but are not QTL *per se*, because genome domains determining BP-controlling mechanisms should be common between rodents and humans.

### (D) *C8QTL2* of DSS rats functionally captures human GWAS genes [9]

(**a**) In contrast to *C8QTL1*, normotensive alleles of *C8QTL2* raise BP *in vivo* by +30% in the difference between 2 parental strains (Fig 1B, Table 1). Among 11 human GWAS genes in the *C8QTL2*-residing region (S1 Table), 2 of them can be in 2 genome blocks separated by a 17 Mb gap and correspond to 2 functional candidates from rodents carrying missense mutations (Fig 1B and Table 1). This implies 2 distinct QTLs in the broad region presently defined to contain *C8QTL2*. This functional evidence has united at least one of human GWAS genes and rodent *C8QTL2*, beyond an overall similarity in BP between the 2 orders of mammals.

(**b**) We next prioritized functional candidate genes based on the existence of missense mutations in them. *Mtnr1b* and *Snx19* coding domains bear the same missense mutations (Table 1) in our rat data base [25] and RGD [26], although we have not singularly cloned and sequenced all their exons. Coding regions of human *MTNR1B* and *SNX19* are highly conserved with those of rats and contain various missense mutations (S4 Table). Rat orthologs for remaining 9 GWAS genes in the *C8QTL2*-residing interval (S1 Table) do not carry missense mutations and thus are not prioritized as functional candidates.

In contrast to the conserved coding domains in *MTNR1B* and *SNX19* between rats and humans, non-coding GWAS SNPs marking them only exist in primates including humans (S3 Table), but are 'knocked out' in rodents (S2 Table). Since BP changed by congenic knock in genetics independently of them, they cannot be responsible for the functionality of *C8QTL2*, because genome domains regulating BP should be shared in both rodents and humans.

### (E) *C7QTL1* of DSS rats may belong to a functional human ortholog of GWAS genes [9]

(**a**) Normotensive alleles of *C7QTL1* lower BP *in vivo* (Fig 1A). Among 21 human GWAS genes in the *C7QTL1*-residing region (S1 Table), 2 of them can be in 2 genome blocks separated by a 1.7 Mb gap (Fig 1B and Table 1). This implies 2 isolated QTLs in the broad region presently known to contain one QTL, *C7QTL1*. This functional evidence has unified, at minimum, one of human GWAS genes and the rodent *C7QTL1*.

(**b**) We next identified potential functional candidate genes based on the existence of missense mutations in them. *Triobp* and *Tnrc6b* coding domains bear missense mutations (Table 1) based on our rat data base [25] and RGD [26]. We then cloned and sequenced the exons suspected to contain the missense mutations. These mutations are confirmed (Table 1). Rat orthologs for remaining 19 GWAS genes in the *C7QTL*-residing interval (S1 Table) do not carry missense mutations and thus are not prioritized as functional candidates at this point.

In contrast to the conserved coding domains between rat and humans, non-coding GWAS SNPs marking them only exist in primates including humans (S3 Table), but are 'knocked out' in rodents (S2 Table). Since BP changed by congenic knock in genetics without them, they cannot be responsible for the function of *C7QTL1*, because of conserved BP-modulating mechanisms between rats and humans.

## Discussion

Primary findings from this work are (**a**) *in vivo* studies and human GWAS have begun to consolidate mechanisms of BP control into a biological framework for a quantitative and polygenic trait. QTL modularity on BP is conserved between divergent mammalian orders such as

rodents and humans (www.timetree.org). (**b**) As a proof of principle, 3 distinct QTLs from inbred DSS rats have functionally captured at least 3 human GWAS genes. Each of them has a major physiological impact on BP. (**c**) The non-coding SNPs marking these 3 QTLs/human GWAS genes are spin-offs of primate evolution autonomously of BP regulation. They mark potential QTLs nearby, rather than QTLs themselves.

## QTL Modularity/non-cumulativity explains not only physiological controls on BP but also their evolutionary conservation

A BP effect for a given GWAS signal in a general population seemed miniscule as it is often fractionated from total BP variance [9]. Total variance gauges a spread of BP in the heterogeneous populations and largely due to environments, not due to QTLs. Since environmental factors are not inherited, the often-referred to 'missing' heritability for BP is not physiologically mechanistic. Missing total variance is not the same as 'missing' heritability. In physiological reality, each of the 3 QTLs of human orthologs alone showed a major effect on BP by function, although BP effects for many more GWAS genes are not known due to a lack of fine resolution from congenic knock ins (Fig 1). Thus, a pertinent functional issue to address is how they can collectively impact on BP, in spite of their functional redundancy.

Several human GWAS genes may be grouped into only 2 epistatic modules by their functions together in physiologically controlling BP. This insight implicates 2 pathways of hypertension pathogeneses and one of them is the M3R signaling pathway [21]. Thus, QTL modularity on BP is conserved between differing orders of mammals as divergent as humans and rodents and proves in principle that physiological pathways regulating BP are shared between humans and rodents.

Since the physiological system of BP control in mammals including rodents and humans has not changed before and since their evolutionary divergence [2], the occurrence of non-coding SNPs uniquely in humans is a coincidence in primate evolution. Such a non-coding GWAS SNP marks the position of a functional QTL nearby, similar to the microsatellite polymorphism [28] signalling *C17QTL1/Chrm3* next to it [21]. Thus, replicating such a GWAS SNP is an epidemiological exercise independently of identifying the QTL next to it that actually affects blood pressure, but carries less prevalent missense mutations.

## Mechanistic and physiological insights into controlling blood pressure as a polygenic trait from QTL modularity

The modularity of QTLs [10, 15] has enlarged the coverage of Mendelism, added a new physiological dimension to polygenic trait genetics, and provided a mechanistic framework for how redundant QTLs act together in regulating mammalian BP including those in humans.

Lately, an 'omnigenic' hypothesis has been expressed to cover GWAS results on various phenotypes including blood pressure [13]. It can even be referred to as an anthropocentric (or human-centered) hypothesis, because non-coding SNPs used in GWAS only exist in humans, not in rodents. It essentially suggests that it is regulations at gene expressions that determine the GWAS SNPs' roles in any phenotypes including BP in humans, contrary to the modularity paradigm [10, 15].

The basic difference between the two is that the modularity concept is based on the functional physiology with verifiable pathogenic mechanisms of hypertension versus statistical epidemiology without a functional physiology and mechanisms. This is because mechanisms and physiology determining a polygenic trait are only an afterthought, not the initial driving force or the starting point in the omnigenic model. These 2 conflicting hypotheses produce different predictions that can be experimentally tested for their physiological relevance on BP. From the

perspective of functional physiology in blood pressure regulation, modularity is supportive, but omnigenicity is not, because of the following.

(a). A gene dose, and by inference, the level of gene expressions are central to the 'omnigenic' hypothesis. However, several lines of experimental evidence have proven that a gene dose is irrelevant to the physiological impact on the hypertension pathogenesis. For example, *Chrm3*$^{+/-}$ has only one functional copy of *Chrm3* and *Chrm3*$^{+/+}$ has 2 copies. Despite this difference in gene copies, blood pressures of both genotypes are the same [21, 22]. Multiple QTLs behave in this gene-dose independence in the functional physiology of BP control [29].

(b). The infinitesimal effect from a single QTL marked by a GWAS SNP is predicted from the 'omnigenic' hypothesis along with a phenotypic gradient in response to the dose of QTLs. If this assumption were physiologically relevant, depleting one such QTL should have a trivial effect on BP. This is not the case. Depleting *Chrm3* alone causes BP to drop more than 50% comparing the full-body *Chrm3*$^{-/-}$ with *Chrm3*$^{+/+}$ [21]. Not only that, BP diminished sharply (e.g. 28%) when normotensive alleles of *Chrm3* is knocked in to the background of hypertensive DSS rats [21].

(c). QTL modularity explains redundancy of human BP QTLs in collectively controlling BP, and a non-cumulative correspondence between QTL numbers and BP effects. Adding more QTL alleles from another epistatic module changed blood pressure in a 'leap', not in a gradual gradient and not in proportion [15]. The 'omnigenic' hypothesis predicted the opposite.

(d). The modularity concept can, whereas the 'omnigenic' hypothesis cannot, support the evolutionary conservation in modules/pathways controlling BP between rodents and humans that are separated in time by 90 million years (timetree.org). The non-coding GWAS SNPs marking human QTLs only began to appear in primates, but do not exist in rodents with whom humans shared same BP-regulating mechanisms. Thus, it is the functional impact of Chrm3 signaling that determines the hypertension pathogenesis, not any of the non-coding rodent SNPs around the Chrm3 codons and the non-existent human GWAS SNP. This is because knocking out the Chrm3 signaling did not touch any of these rodent SNPs, yet BP changed. The Chrm3 signaling is conserved between humans and rodents.

Thus, the human BP physiology is not peculiarly different from that of rodents, nor is it more complicated. It's simply more difficult to dissect and distinguish than that of inbred rodents due to experimental limitations on revealing pathogenic mechanisms by function. In the vast scheme of mammalian BP-regulating mechanisms, rodents and humans are simply tiny branches on an evolutionary tree supported by a central mammalian 'trunk'.

(e). The involvement of certain QTLs [21] starts at embryogenesis, before the onset of adult BP physiology. The 'omnigenic hypothesis' does not seem to explain how 'regulations at gene expressions' at embryogenesis could affect adult BP. In contrast, the modularity concept explains that a pathway is involved in BP control and a pathway can temporally begin at embryogenesis [21].

(f). In summary, the QTL modularity concept is supported by uncontested physiological evidence as a signaling pathway, and is broader and more profound in our understandings of mechanisms of BP control than the 'omnigenic' hypothesis, which has little functional support for any physiological role on BP.

## Dissociating 'common' GWAS non-coding variants from closely-linked 'rare' coding mutations

Natural 'knock outs' of non-coding GWAS SNPs in our current work showed that they do not have a physiological impact on blood pressure. They might potentially have functions in

modulating gene expressions, epigenetics and/or even be eQTLs, but they are not involved in physiological regulations of BP in mammals including humans.

The use of GWAS is, nonetheless, useful in labeling the vicinity of functional QTLs close by [9]. We have discussed in depth the issue of a marker being separable from a functional QTL next to it, and the ambiguity of statistical confidence interval in the LD application in our previous publication [14]. This topic will not be repeated here.

Statistical incertitude may lead to a detected signal for a QTL being false negative or false positive [7], to which a SNP's prevalence is immaterial. A proof by function is the only yardstick in assessing it. Not detecting *C7QTL1* and *C7QTL2* is an example of statistical false negativity [23] and was rectified only by a functional proof in changing BP in congenic knock in genetics (Fig 1A). A lack of functional impact on BP proved that a statistical detection of a QTL near *inducible nitric oxide synthase* was false positive [30].

## Dissociation of blood pressure with vaso-dilation mediated by M3R encoded by *Chrm3*

Recently, an endothelial cell-targeted *Chrm3* knock out in mice showed that blood pressure was not affected [31]. Their results are consistent with those in the full-body knockout of *Chrm3* of rat and mice in that vasodilation mediated by M3R is diminished [21, 32]. Despite an impaired vasodilation, blood pressure in our full-body *Chrm3* knock out rats was lowered, not raised [21]. This paradox has created a physiological puzzle [33] that blood pressure is not regulated by vasodilation *per se* mediated by the M3R-mediated signaling.

This separation suggests the following in the genetics of hypertension in either humans or animal models. Unknowing a precise mechanism in blood pressure regulation, the pivotal starting point in proving a QTL has to be its functional impact on blood pressure itself *in vivo*. Any 'intermediate phenotype' connected to blood pressure control cannot substitute this indispensable functional proof. This proof will then drive our mechanistic understandings of the gene, irrespective of our intuitive and known ideas on how blood pressure regulations should be mediated. This function-mechanism consideration can be extrapolated to any other quantitative and polygenic trait in addition to blood pressure, especially when obtained from GWAS.

Concerning M3R, 'the absence of M3R does not mean an endothelial dysfunction, only that the M3R-mediated pathway is inactivated. Many M3R-independent pathways in the endothelium remain active and likely important for BP regulation. Since BP decreases *in vivo* in the (full-body) *Chrm3*-nulls, even if endothelium-derived acetylcholine could contribute to endothelium-dependent dilation, it is likely to be compensated *in vivo* by other mechanisms such as improved renal and cardiac activities' [22].

## Pathogenic pathways of hypertension inferred from the molecular bases of QTLs

Since the 4 QTLs in questions (Fig 1) have not been molecularly identified, their roles in BP control are tentative and inferred only from the functional candidate genes with missense mutations representing them.

In epistatic module 2/pathway 2 as muscarinic cholinergic receptor 3 [21, 22], MTNR1B is a G-protein-coupled melatonin receptor primarily expressed in the brain retina. Knocking it out showed no detectable phenotype [34], although blood pressure was not measured SNX19 seems to function in pancreatic β cells [35] and ubiquitously expressed [36].

In epistatic module 1/pathway 1, 3 functional candidate genes, *Ulk3*, *Cyp1a2*, and *Lman1l*, are present in block 1 in the *C8QTL1*-lodging region (Fig 1B). ULK3 is a serine/threonine

kinase involved in embryonic hedgehog pathway [37], although its tissue distribution is mostly in skin and small intestine. *CYP1A2* is a cytochrome P450 enzyme specifically expressed in liver [38] and involved in eliminating exogenous chemicals. LMAN1l is a mannose-binding and lectin-like protein involved in organelle trafficking [39] and primarily expressed in prostate [40]. A siRNA delivery of *Ulk3* or *Cyp1a2* did not alter blood pressure [41]. *Lman1l* was included in a mouse knock out library [42], but no phenotype was detailed. *Cyp1a2* null mice showed a deficient drug metabolism [38]. *CCDC33* is primarily expressed in testes [43], and appeared to be not vital for development [44]. TRIOBP controls cytoskeleton organization and is involved in hearing [45]. Tnrc6b is a protein with repeated glycine/tryptophan residues, may play a role in gene silencing by small interfering-RNAs and microRNAs, and is semi-required for development [46].

## Caveats and limitations

First, although human GWAS genes with missense mutations do not genetically prove by themselves to be the QTLs in questions, they provide entry points towards probable pathways underlying the function of each QTL. They are molecular targets for designing viable gene-specific experiments in validating their functions on blood pressure. Unlike non-existence of non-coding GWAS SNPs (S2 Table), the coding mutations can be tested in rodents by function.

Second, structural mutations are not necessarily the only molecular bases that can affect a pathway in question. Although bearing no missense mutation, *CSK* encodes a c-src tyrosine kinase, is located between *Cyp1a2* and *Lman1l* and a candidate gene for a QTL. A *Csk* knock down and *Csk*$^{+/-}$ showed a decrease in blood pressure [41]. This directionality is the opposite of what expected from the human non-coding GWAS SNP, rs1378942 [9], which is naturally 'knocked out' in mice and had no effect on BP [41].

*Atp2b1* in the *C7QTL2*-residing interval (Fig 1A) carries no missense mutation. However, an *Atp2b1* knock down has shown an increase in blood pressure [47]. A vascular smooth muscle-targeted *Atp2b1* knock out showed an increase in blood pressure [48] as well as in heterozygotes of a full-body knock out [49]. The fact that *Atp1b1* and *Csk* nulls are not viable indicates that their functional involvement in BP control may begin in embryogenesis [50].

In conclusion, both *in vivo* rodent studies and human GWAS have revealed common QTLs. Each of them has a major functional impact on BP, even though non-coding GWAS SNPs next to them have no such functional effects. Missense mutations of specific genes qualify them to be functional candidates for certain QTLs. The shared epistatic modularity among human and rodent QTLs suggests that they may function in a common pathway evolutionarily conserved, and each is involved in a different step in the pathogenic pathway for polygenic hypertension.

## Supporting information

**S1 Checklist. ARRIVE guidelines checklist.**
(PDF)

**S1 Table. Rat QTLs and genes.**
(DOCX)

**S2 Table. A survey of sequence homologies between humans and the rats for non-coding GWAS SNPs.**
(DOCX)

**S3 Table. Survey of non-coding GWAS SNP conservations/homology during primate evolution.**
(DOCX)

**S4 Table. Coding mutation alignments for rat and human genes.**
(DOCX)

## Author Contributions

**Conceptualization:** Alan Y. Deng.

**Data curation:** Annie Ménard.

**Formal analysis:** Alan Y. Deng.

**Funding acquisition:** Alan Y. Deng.

**Investigation:** Alan Y. Deng.

**Methodology:** Annie Ménard.

**Supervision:** Alan Y. Deng.

**Validation:** Alan Y. Deng.

**Writing – original draft:** Alan Y. Deng.

**Writing – review & editing:** Alan Y. Deng.

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
