## [Decision Letter · Decision Letter 0]

9 Mar 2020

PONE-D-19-35461

Conserved mammalian modularity of quantitative trait loci revealed human functional orthologs in blood pressure control

PLOS ONE

Dear Dr. Deng,

Thank you for submitting your manuscript to PLOS ONE. After careful consideration, we feel that it has merit but does not fully meet PLOS ONE’s publication criteria as it currently stands. Therefore, we invite you to submit a revised version of the manuscript that addresses the points raised during the review process.

The reviewers agree that the manuscript is just partially sound and arise some doubts on the novelty of the manuscript with respect to the previous contributions by the authors themselves and the related literature. Moreover, the rigour with which the statistical analysis has been performed can be improved, and the insights into the behaviour of blood pressure control should be better clarified.

We would appreciate receiving your revised manuscript by Apr 23 2020 11:59PM. To enhance the reproducibility of your results, we recommend that if applicable you deposit your laboratory protocols in protocols.io, where a protocol can be assigned its own identifier (DOI) such that it can be cited independently in the future. For instructions see: http://journals.plos.org/plosone/s/submission-guidelines#loc-laboratory-protocols

We look forward to receiving your revised manuscript.

Kind regards,

Alessandro Borri

Academic Editor

PLOS ONE

Journal Requirements:

2. Please complete and submit a copy of the ARRIVE Guidelines checklist, a document that aims to improve experimental reporting and reproducibility of animal studies for purposes of post-publication data analysis and reproducibility: https://www.nc3rs.org.uk/arrive-guidelines. Please include your completed checklist as a Supporting Information file. Note that if your paper is accepted for publication, this checklist will be published as part of your article.

Specifically, please ensure that you revise your methods section to include the method of euthanasia, as well as how frequently the condition of the animals was monitored.

Reviewers' comments:

Reviewer's Responses to Questions

**Comments to the Author**

1. Is the manuscript technically sound, and do the data support the conclusions?

Reviewer #1: Partly

Reviewer #2: Partly

2. Has the statistical analysis been performed appropriately and rigorously? 

Reviewer #1: I Don't Know

Reviewer #2: No

3. Have the authors made all data underlying the findings in their manuscript fully available?

Reviewer #1: Yes

Reviewer #2: Yes

4. Is the manuscript presented in an intelligible fashion and written in standard English?

Reviewer #1: No

Reviewer #2: Yes

5. Review Comments to the Author

Reviewer #1: TO THE AUTHORS:

PLOS One; PONE-D-19-35461

Title: Conserved mammalian modularity of quantitative trait loci revealed human functional orthologs in blood pressure control

Deng & Menard seek to publish a follow-up study on the genetic underpinnings of hypertension that builds from their recently published work (e.g., Deng & Menard, PMID: 31584514). In particular, the authors are seeking to reconcile findings of human epidemiology with genetic mechanisms of rodent models with tools that utilize a modularity (vs. an “omnigenic” regulation) approach. The manuscript demonstrates knowledge of the nuances involved in using respective hypertension models (humans vs. rats) and interpretation of the resulting data thereof. Also, they endeavor to harness scientific reductionism in their general approach (e.g., rat to human orthologs) to help open up more paths of translation towards effective blood pressure control in human subjects. See my comments below.

(1) General: As presented, it is not readily apparent that this manuscript is a significantly new and distinguished contribution relative to previous efforts (e.g., Deng & Menard 2019, PMID: 31584514). Although objectives/accomplishments are listed (Introduction, Lines 86-90), there needs to be a more clearly presented case for unfulfilled questions left over from previous efforts and how this study addressed them.

(2) General: The authors could improve the manuscript throughout by considering and providing more of a balanced discussion of their findings in light of the literature. Also, as appropriate for scientific efforts and reports, statements of data should be limited to being supportive or suggestive of their conclusions rather than the only “right” or “valid” answer(s) vs. the broader peer-reviewed literature.

(3) General: Toggling various genes on and off or mutating them in a global manner with fortuitous consequences on blood pressure does not necessarily illuminate their individual (or interactive) roles in the grand scheme of blood pressure control or make them practical for understanding/treating hypertension. As an example, Chrm3 deletion specific to endothelial cells (a cell type central to determining vascular resistance and thus, blood pressure) does not appear to significantly alter peripheral vascular resistance, arterial tone, arterial blood pressure, or cardiac function (Rhoden et al 2019, PMID: 30716211). In tandem, this same study reported that the Chrm3 endothelium-specific knockout animal also contained reduced expression of mRNA transcripts for isoforms 1, 2, 4, & 5 (1, 3, 5 are Gq-type and 2 & 4 are Gi-type) by >40% in a vascular endothelium specific knockout of Chrm3 vs. wild-type. The potential for heteromeric interaction among isoforms as integral membrane proteins can not be excluded either. Again, no one study makes for a complete “right” answer but well-evidenced possibilities (and other layers of regulation supporting physiological function) outside of and/or complementary to the authors’ approaches and hypotheses should be considered as well.

(4) Although published previously, the authors should still at least define and specify animal models (DSS, biological sex, etc.) up front in the Animals section of Materials and Methods.

Reviewer #2: The authors have proposed an extension of their previous studies [15][16] to PLoS ONE. Since it is an extension, careful content delineation from the previous studies is expected. Unfortunately, it appears to me that the extension is weak and vague. I do not see any novel or sound insight into the study.

1. Statistical analyses were not rigorously performed in the study.

2. There is not any new data or algorithm in the study.

3. The insights into BP are vague and speculative.

4. The writing is pretty confusing with lots of specific but unnecessary terminologies.

6. PLOS authors have the option to publish the peer review history of their article (what does this mean?). If published, this will include your full peer review and any attached files.

Reviewer #1: Yes: Erik Behringer

Reviewer #2: No

---

## [Author Response · Author response to Decision Letter 0]

26 Mar 2020

Reviewer #1: Deng & Menard seek to publish a follow-up study on the genetic underpinnings of hypertension that builds from their recently published work (e.g., Deng & Menard, PMID: 31584514). In particular, the authors are seeking to reconcile findings of human epidemiology with genetic mechanisms of rodent models with tools that utilize a modularity (vs. an “omnigenic” regulation) approach. The manuscript demonstrates knowledge of the nuances involved in using respective hypertension models (humans vs. rats) and interpretation of the resulting data thereof. Also, they endeavor to harness scientific reductionism in their general approach (e.g., rat to human orthologs) to help open up more paths of translation towards effective blood pressure control in human subjects. See my comments below.

Response to Reviewer #1 summary comment: Your summary is to the point. We agree.

(1) General: As presented, it is not readily apparent that this manuscript is a significantly new and distinguished contribution relative to previous efforts (e.g., Deng & Menard 2019, PMID: 31584514). Although objectives/accomplishments are listed (Introduction, Lines 86-90), there needs to be a more clearly presented case for unfulfilled questions left over from previous efforts and how this study addressed them.

Response to Reviewer #1 comment (1): You are right. We have added a section in the introduction to present unfulfilled questions left over and how the current study planned to address them as follows.

Nevertheless, this initial work only covered a small section of the genome. Other rat chromosome regions that contain orthologs of human GWAS genes have not been studied. 

(2) General: The authors could improve the manuscript throughout by considering and providing more of a balanced discussion of their findings in light of the literature. Also, as appropriate for scientific efforts and reports, statements of data should be limited to being supportive or suggestive of their conclusions rather than the only “right” or “valid” answer(s) vs. the broader peer-reviewed literature.

Response to Reviewer #1 comment (2): Thanks for pointing out the latest publication by Rhoden et al 2019, PMID: 30716211. In discussions, the results from this paper have been discussed to balance our results. A broader peer-reviewed literature is mostly presented in the Introduction. We have tried to tone down statements as you recommended throughout the manuscript as much as we can.

(3) General: Toggling various genes on and off or mutating them in a global manner with fortuitous consequences on blood pressure does not necessarily illuminate their individual (or interactive) roles in the grand scheme of blood pressure control or make them practical for understanding/treating hypertension. As an example, Chrm3 deletion specific to endothelial cells (a cell type central to determining vascular resistance and thus, blood pressure) does not appear to significantly alter peripheral vascular resistance, arterial tone, arterial blood pressure, or cardiac function (Rhoden et al 2019, PMID: 30716211). In tandem, this same study reported that the Chrm3 endothelium-specific knockout animal also contained reduced expression of mRNA transcripts for isoforms 1, 2, 4, & 5 (1, 3, 5 are Gq-type and 2 & 4 are Gi-type) by >40% in a vascular endothelium specific knockout of Chrm3 vs. wild-type. The potential for heteromeric interaction among isoforms as integral membrane proteins can not be excluded either. Again, no one study makes for a complete “right” answer but well-evidenced possibilities (and other layers of regulation supporting physiological function) outside of and/or complementary to the authors’ approaches and hypotheses should be considered as well.

Response to Reviewer #1 comment (3): We’d like to thank you again for pointing out the last publication on a Chrm3 conditional knock out (Rhoden et al 2019, PMID: 30716211). Their results are consistent with those in our full-body knockout of Chrm3 in that vasodilation mediated by its protein product is diminished, and acts independently of blood pressure control. 

Despite an impaired vasodilation, blood pressure in our full-body Chrm3 knock out was lowered, not raised (Deng et al 2018, PMID: 30354759), whereas depleting Chrm3 from endothelial cells by Rhoden and coworkers did not change blood pressure. This paradox has created a physiological puzzle (Cowley 2018, PMID: 30354773). That means that blood pressure is not regulated by vasodilation per se mediated by the Chrm3-mediated pathway in the endothelial cells. 

This separation suggests the following in the genetics of hypertension in either humans or animal models. Unknowing a precise mechanism in blood pressure regulation, the pivotal starting point in proving a QTL has to be its functional impact on blood pressure itself in vivo. Any ‘intermediate phenotype’ connected to blood pressure control cannot substitute this indispensable functional proof. This proof will then drive our mechanistic understandings of the gene, irrespective of our intuitive and known ideas on how blood pressure regulations should be mediated. This function-mechanism consideration can be extrapolated to any other quantitative and polygenic trait in addition to blood pressure, especially on GWAS.

Concerning Chrm3 encoding M3R, it is further discussed in (Deng et al 2019, 10.1016/j.cjca.2018.12.029’). ‘The hypertension pathophysiology directed by M3R is largely dissociated from the M3R-mediated vaso-relaxation. However, the absence of M3R does not mean an endothelial dysfunction, only that the M3R-mediated pathway is inactivated. Many M3R-independent pathways in the endothelium remain active and likely important for BP regulation. Since BP decreases in vivo in the (full-body) Chrm3-nulls, even if endothelium-derived acetylcholine could contribute to endothelium-dependent dilation, it is likely to be compensated in vivo by other mechanisms such as improved renal and cardiac activities’.

A new section has been added in the Discussion.

(4) Although published previously, the authors should still at least define and specify animal models (DSS, biological sex, etc.) up front in the Animals section of Materials and Methods.

Response to Reviewer #1 comment (4): We agree with you. Some information as you requested has been added. Our basic animal model is inbred Dahl salt-sensitive (DSS) rats. Congenic strains were made by replacing distinct chromosome regions of DSS by those of inbred normotensive rats.

Reviewer #2: The authors have proposed an extension of their previous studies [15][16] to PLoS ONE. Since it is an extension, careful content delineation from the previous studies is expected. Unfortunately, it appears to me that the extension is weak and vague. I do not see any novel or sound insight into the study.

Response to Reviewer #2 general comment: We have delineated novelty in the revision. Our first-phase studies were limited to a few rat QTLs of human GWAS orthologs. They provided a proof of principle study. However, there are a lot more rat QTLs that correspond to additional human GWAS genes. We are reporting our studies on them here in this manuscript.

A sentence was added to the beginning of last section in Introduction as follows: 

‘Nevertheless, this initial work only covered a small section of the genome. Other rat chromosome regions that contain orthologs of human GWAS genes have not been studied’. We hope now the novelty is clearer than our previous version.

1. Statistical analyses were not rigorously performed in the study.

Response to Reviewer #2 comment (1): Our statistical analyses have been reported extensively in our previous studies. We did not put them in our original submission to Plos One. Now in addressing your concerns, we have added statistics to the end of Material and Method section as follows: 

Repeated measures' analysis of variance (ANOVA) followed by the Dunnett’s test was used to compare the significance in a difference or a lack of it in a blood pressure component between a congenic strain and the DSS parental strain. The Dunnett correction takes into account multiple comparisons as well as sample sizes among the comparing groups. In the analysis, a BP 

component was compared on each day for the period of measurement among the strains. 

Because blood pressures of rats were measured continuously for 2 weeks with one reading at every 2 minutes, the numbers given at the bottom of Fig. 1 represent only averaged values of mean arterial pressures for a strain during 2 weeks. A Dunnett value including “<” given in each comparison between congenic and DSS strains was the most conservative P value among all the days of comparisons.

In case that you may suggest a genome-wide statistics, it’s not relevant to comparing a congenic trait with the DSS parental rats.

2. There is not any new data or algorithm in the study.

Response to Reviewer #2 comment (2): Please see our responses to your comment (1) above.

3. The insights into BP are vague and speculative.

Response to Reviewer #2 comment (3): In our current study, we have provided the physiological evidence, for the first time, that certain human GWAS genes functionally affect blood pressure by rat QTL proxies. As you may know, GWAS results are probabilistic and statistical. No functional data can be obtained regarding the blood pressure physiology. The next phase of studies will be to molecularly identify the QTL/human GWAS gene that is involved in hypertension pathogenesis. Mechanistic insights will follow. We suppose what you meant by ‘vague and speculative’ is that a specific gene has not been molecularly identified to be responsible for hypertension pathogenesis. These are limitations of current studies that are recognized in Caveats and limitations section of the manuscript.

4. The writing is pretty confusing with lots of specific but unnecessary terminologies.

Response to Reviewer #2 comment (4): We attempted at minimizing specific terminologies in the revision. However, some terminologies are necessary to convey the scientific message, such as congenic strains, congenic knock in genetics, GWAS, SNPs, QTLs, and gene names CSK etc.

see cover letter for more

---

## [Decision Letter · Decision Letter 1]

1 May 2020

PONE-D-19-35461R1

Conserved mammalian modularity of quantitative trait loci revealed human functional orthologs in blood pressure control

PLOS ONE

Dear Dr. Deng,

Thank you for submitting your manuscript to PLOS ONE. After careful consideration, we feel that it has merit but does not fully meet PLOS ONE’s publication criteria as it currently stands. Therefore, we invite you to submit a revised version of the manuscript that addresses the points raised during the review process.

The Reviewers of the previous round were in total disagreement in their assessments, so I needed to involve two additional reviewers, whose evaluation is somehow positive but not sufficiently deep in some points. One first-stage reviewer, indeed, found that your reviewing effort at the previous round was insufficient.

The main issue to be addressed is the novelty of this contribution with respect to the previous work by the authors, in particular the recent paper [14], with which the discussion has much overlap. Furthermore, please check and update references and address all the remaining concerns according to the Reviewers' comments.

We would appreciate receiving your revised manuscript by Jun 15 2020 11:59PM. To enhance the reproducibility of your results, we recommend that if applicable you deposit your laboratory protocols in protocols.io, where a protocol can be assigned its own identifier (DOI) such that it can be cited independently in the future. For instructions see: http://journals.plos.org/plosone/s/submission-guidelines#loc-laboratory-protocols

We look forward to receiving your revised manuscript.

Kind regards,

Alessandro Borri

Academic Editor

PLOS ONE

Reviewers' comments:

Reviewer's Responses to Questions

**Comments to the Author**

1. If the authors have adequately addressed your comments raised in a previous round of review and you feel that this manuscript is now acceptable for publication, you may indicate that here to bypass the “Comments to the Author” section, enter your conflict of interest statement in the “Confidential to Editor” section, and submit your "Accept" recommendation.

Reviewer #1: (No Response)

Reviewer #2: (No Response)

Reviewer #3: All comments have been addressed

Reviewer #4: (No Response)

2. Is the manuscript technically sound, and do the data support the conclusions?

Reviewer #1: Partly

Reviewer #2: (No Response)

Reviewer #3: Yes

Reviewer #4: Yes

3. Has the statistical analysis been performed appropriately and rigorously? 

Reviewer #1: I Don't Know

Reviewer #2: (No Response)

Reviewer #3: Yes

Reviewer #4: Yes

4. Have the authors made all data underlying the findings in their manuscript fully available?

Reviewer #1: Yes

Reviewer #2: (No Response)

Reviewer #3: Yes

Reviewer #4: Yes

5. Is the manuscript presented in an intelligible fashion and written in standard English?

Reviewer #1: No

Reviewer #2: (No Response)

Reviewer #3: Yes

Reviewer #4: Yes

6. Review Comments to the Author

Reviewer #1: TO THE AUTHORS:

PLOS One; PONE-D-19-35461.R1

Title: Conserved mammalian modularity of quantitative trait loci revealed human functional orthologs in blood pressure control

Although some of my concerns have been sufficiently addressed while bringing up exciting questions for further study (regulation of endothelial-dependent vascular tone vs. regulation of systemic blood pressure), the revised presentation of the original manuscript appears to be a very modest effort. Remaining concerns are indicated below.

(1) General: There is a remaining concern that the authors don’t adequately present how the current manuscript is a new and distinguished contribution relative to previous efforts. The added statement (along with the rest of the last paragraph) in the Introduction does not add any information. The reader will like to know what a “small section of the genome” means and why the authors logically stopped there the first time around as a study for publication. Likewise, the reader will want to see examples or classifications of what “other rat chromosome regions” corresponding to human GWAS genes are left to study and why. The first summary paragraph in the Discussion is not altogether convincing of a new contribution either. These boundaries among studies and respective manuscripts need to be clear, especially in light of apparently new contributions (e.g., PMID: 31584514) since original review of the current manuscript.

(2) General: Statements and supporting references need to be checked throughout the manuscript for validity. For example, there is a statement in the Introduction (Lines 65-66; “This difficulty…known to affect blood pressure) with citation of references that are approximately 7 years old now.

Reviewer #2: Please use the space provided to explain your answers to the questions above. You may also include additional comments for the author, including concerns about dual publication, research ethics, or publication ethics. (Please upload your review as an attachment if it exceeds 20,000 characters) (Limit 100 to 20000 Characters)

Reviewer #3: I congratulate Deng & Menard on their revision, and scientific efforts pertaining to the genetics of HTN. In this revision, they have addressed all comments. Their data is complex yet significant, and adds to their existing contribution to the literature. Higher resolution figures should be submitted.

Reviewer #4: Deng and Menard present results of their continued work aiming to identify functional genetic variants related to blood pressure. In this particular study, they report on 3 QTLs present in DSS rats that map to numerous GWAS human genes linked to blood pressure. The authors also make the point that non-coding SNPs in GWAS act as markers of nearby QTLs rather than being QTLs themselves. Since functional genomics research lags behind the rate of statistically-based human GWAS, this work represents important findings and considerations. However, as noted by the previous reviewers, these authors recently have published a similar paper with the same overall messages. While new QTLs are discussed in the present manuscript, the Discussion of this manuscript has a degree of overlap with the Discussion of the J Hypert 2020 article (citation #14).

1. Although the authors modified the Introduction to state that the present analysis varied from the previous paper due to inclusion of other rat chromosome regions, I was left wondering if with this additional study a comprehensive set of the rat genome has now been evaluated by this group? How much more of the rat genome did the authors cover with this extension to the original work?

2. Line 102 "They are basically the same as documented previously [15]." First, I believe the authors meant to cite reference 14 (not 15). Please verify. Second, this sentence would benefit from being rewritten. Stating that the methods were "basically the same" leaves one wondering if something was different and if so what? Please revise to provide clarity on how the methods differed, and if they did not differ, indicate that the methods were the same.

3. The Discussion section is overly lengthly and has overlap with the original paper (citation 14). Particularly the material on noncoding GWAS SNPs serving as markers for nearby QTLs, a point that was discussed already at length in citation #14. In this manuscript, I very much appreciated the thoughtful Discussion of missing heritability in GWAS and the omnigenic hypothesis. The Discussion needs to be condensed and so reducing content that is similar to the previous publication would shorten the section.

7. PLOS authors have the option to publish the peer review history of their article (what does this mean?). If published, this will include your full peer review and any attached files.

Reviewer #1: No

Reviewer #2: No

Reviewer #3: No

Reviewer #4: Yes: Daiva Nielsen

---

## [Author Response · Author response to Decision Letter 1]

4 May 2020

Reviewer #1: Although some of my concerns have been sufficiently addressed while bringing up exciting questions for further study (regulation of endothelial-dependent vascular tone vs. regulation of systemic blood pressure), the revised presentation of the original manuscript appears to be a very modest effort. Remaining concerns are indicated below.

 Response to Reviewer #1 summary comment: We’d like thank you for recognizing our responses to your previous questions. It seems that you agree with scientific merits of our current work such as a major physiological effect from each QTL and a redundancy of multiple QTLs in their effects on blood pressure, modularity conservation between humans and rats, separation of SNP markers originating from primate evolution from functional genes nearby on blood pressure .

 We are addressing your logistic concerns below. Also, we have added new phrases in Introduction as ‘Here, we aimed at expanding the scope of rat QTL coverage to capture additional human GWAS genes in progressive stages, since we are a single investigator-driven lab. In the process, our new results have validated the reproducibility of our previous findings. We focused on distinct regions on DSS rat Chromosomes 7 and 8 that contain previously-unexplored rat blood pressure QTLs and human GWAS gene orthologs’. Hopefully, you are satisfied with them

Reviewer #1 commment (1): There is a remaining concern that the authors don’t adequately present how the current manuscript is a new and distinguished contribution relative to previous efforts. The added statement (along with the rest of the last paragraph) in the Introduction does not add any information. The reader will like to know what a “small section of the genome” means and why the authors logically stopped there the first time around as a study for publication. Likewise, the reader will want to see examples or classifications of what “other rat chromosome regions” corresponding to human GWAS genes are left to study and why. The first summary paragraph in the Discussion is not altogether convincing of a new contribution either. These boundaries among studies and respective manuscripts need to be clear, especially in light of apparently new contributions (e.g., PMID: 31584514) since original review of the current manuscript.

 Response to Reviewer #1 comment (1): We suspect that you were thinking of our current work as a partial overlap with our previous published work. You may wonder why we are submitting similar work on the genetics of other genome sections as a separate paper, instead of having all data put together as one publication. You wondered what justifications there is in presenting new data, but with similar conclusions.

 You seem to think of our effort from the logistic stand point of view of GWASs, where hundreds of investigators were involved in consortia with thousands of supporting staff and a huge combined budget. That’s why these GWASs were able to study associations from the whole genomes from multiple populations simultaneously, and published them as such. For example, reference 8 was able to present data on the entire genome, instead of portions of them. In contrast to this consortium-based research, our studies are individual-investigator based with one part-time research assistant supported by a tiny private fund. You may have noticed the authorship of 2 persons on the current manuscript and our previous publication.

 Faced with the reality of limited resources, we have to comprehensively study QTLs from the entire rat genome corresponding to all human GWAS genes in progressive stages, and fragmented these studies over a long period of time as resources became available. At each stage, we focused on a few QTLs at a time, produced results only on these few QTLs as our resources allowed, analyzed them and sent them for publication. Once the validity of this limited work is peer-reviewed as legitimate and sound, we moved on to the next stage, once again with our very limited resources, to other regions of QTLs, only a few QTLs this second time. The result of second-stage studies used the same method. From this new work, we reproduced previous findings in different sections of the rat genome, and confirmed our previous conclusion. Another stage of investigations was repeated, so on and so forth. 

 As far as we know, our research lab with only 2 persons is the only one in the world doing this kind of work presented in the current manuscript. That is why the reproducibility of our findings is crucial to make our conclusions scientifically valid, as least in our hands. Unfortunately, there is no consortium on the physiology of QTLs in animal models at the present. Most of animal genetic researchers we know have moved on to ‘greener pastures’ of human GWAS. 

 As you may know, the scope of Plos One is to publish valid findings in science with sound methods. In our current manuscript, we do not claim to report the first finding in biologically capturing human GWAS genes with rat orthologs, which was reported in our previous publication (PMID: 31584514). Nevertheless, our current findings on different sections of rat genomes are methodologically sound, and novel in functionally capturing new human GWAS genes, which are different from our previous published results. This reproducibility and confirmation on our previous conclusions are necessary to move the field forward. Newly-identified candidates for different GWAS genes provide new entry point for further gene-targeting studies. Novel mechanistic insights of hypertension pathogenesis will follow.

 You may be aware that even GWASs on blood pressure in human populations were published by same investigators in progressive stages in the past 11 years in high-impact journals such as Nature and Nature Genetics. Each of these incremental publications increased size and ethnicity, decreased minor allele frequencies of study populations from previous one, and also addressed the replicability issue in epidemiology. 

 In comparison, our physiological and functional studies of blood pressure QTLs do not need to increase the number of study subjects to be scientifically valid and to achieve sufficient power. This is because our hypertensive rat model and congenic knock in strains derived from it are inbred and homogeneous. They have been studied under a uniform environment. However, we do need to address the reproducibility issue. We have done so by analyzing different QTLs corresponding to different GWAS genes. As a result, our understanding of mechanisms of hypertension pathogenesis may have been improved with newly identified gene candidates responsible for different QTLs.

 While we understand your comments on boundaries of results, we also like to bring your attention to our logistics and budgetary restraint of doing research in a 1-investigator-plus-one assistant lab. Just to let you know, we did not divide our results into separate publications, but rather we conducted and reported our progressive search from different stages and in an evolutionary time table. Based on our current work, we are planning further research, once again, in a small scale, in the future. We’ll present our future results as they come. We’d appreciate your evaluation when the time comes.

(2) General: Statements and supporting references need to be checked throughout the manuscript for validity. For example, there is a statement in the Introduction (Lines 65-66; “This difficulty…known to affect blood pressure) with citation of references that are approximately 7 years old now.

Response to Reviewer #1 comment (1): Yes, the reference is valid and the first GWAS reported on hypertension.

Reviewer #2: Please use the space provided to explain your answers to the questions above. You may also include additional comments for the author, including concerns about dual publication, research ethics, or publication ethics. (Please upload your review as an attachment if it exceeds 20,000 characters) (Limit 100 to 20000 Characters)

Response to Reviewer #2: There is no comments to respond to.

Reviewer #3: I congratulate Deng & Menard on their revision, and scientific efforts pertaining to the genetics of HTN. In this revision, they have addressed all comments. Their data is complex yet significant, and adds to their existing contribution to the literature. Higher resolution figures should be submitted.

Response to Reviewer #3 comment: Thank you for your appreciation. We have submitted our original figures in revision. Hopefully, the resolution is better.

Reviewer #4: Deng and Menard present results of their continued work aiming to identify functional genetic variants related to blood pressure. In this particular study, they report on 3 QTLs present in DSS rats that map to numerous GWAS human genes linked to blood pressure. The authors also make the point that non-coding SNPs in GWAS act as markers of nearby QTLs rather than being QTLs themselves. Since functional genomics research lags behind the rate of statistically-based human GWAS, this work represents important findings and considerations. However, as noted by the previous reviewers, these authors recently have published a similar paper with the same overall messages. While new QTLs are discussed in the present manuscript, the Discussion of this manuscript has a degree of overlap with the Discussion of the J Hypert 2020 article (citation #14).

 Response to Reviewer #4 summary comment: We thank you for your appreciation of our science in functional genomics research. The following will answer your concerns.

Reviewer #4 comment (1). Although the authors modified the Introduction to state that the present analysis varied from the previous paper due to inclusion of other rat chromosome regions, I was left wondering if with this additional study a comprehensive set of the rat genome has now been evaluated by this group? How much more of the rat genome did the authors cover with this extension to the original work?

 Response to Reviewer #4 comment (1): We suspect that you were thinking of our current work as a partial overlap with our previous published work. You may wonder why we are submitting similar work on the genetics of other genome sections as a separate paper, instead of having all data put together as one publication. You wondered what justifications there is in presenting new data, but with similar conclusions.

 You seem to think of our effort from the logistic stand point of view of GWASs, where hundreds of investigators were involved in consortia with thousands of supporting staff and a huge combined budget. That’s why these GWASs were able to study associations from the whole genomes from multiple populations simultaneously, and published them as such. For example, reference 8 was able to present data on the entire genome, instead of portions of them. In contrast to this consortium-based research, our studies are individual-investigator based with one part-time research assistant supported by a tiny private fund. You may have noticed the authorship of 2 persons on the current manuscript and our previous publication.

 Faced with the reality of limited resources, we have to comprehensively study QTLs from the entire rat genome corresponding to all human GWAS genes in progressive stages, and fragmented these studies over a long period of time as resources became available. At each stage, we focused on a few QTLs at a time, produced results only on these few QTLs as our resources allowed, analyzed them and sent them for publication. Once the validity of this limited work is peer-reviewed as legitimate and sound, we moved on to the next stage, once again with our very limited resources, to other regions of QTLs, only a few QTLs this second time. The result of second-stage studies used the same method. From this new work, we reproduced previous findings in different sections of the rat genome, and confirmed our previous conclusion. Another stage of investigations was repeated, so on and so forth. 

 As far as we know, our research lab with only 2 persons is the only one in the world doing this kind of work presented in the current manuscript. That is why the reproducibility of our findings is crucial to make our conclusions scientifically valid, as least in our hands. Unfortunately, there is no consortium on the physiology of QTLs in animal models at the present. Most of animal genetic researchers we know have moved on to ‘greener pastures’ of human GWAS. 

 As you may know, the scope of Plos One is to publish valid findings in science with sound methods. In our current manuscript, we do not claim to report the first finding in biologically capturing human GWAS genes with rat orthologs, which was reported in our previous publication (PMID: 31584514). Nevertheless, our current findings on different sections of rat genomes are methodologically sound, and novel in functionally capturing new human GWAS genes, which are different from our previous published results. This reproducibility and confirmation on our previous conclusions are necessary to move the field forward. Newly-identified candidates for different GWAS genes provide new entry point for further gene-targeting studies. Novel mechanistic insights of hypertension pathogenesis will follow.

 You may be aware that even GWASs on blood pressure in human populations were published by same investigators in progressive stages in the past 11 years in high-impact journals such as Nature and Nature Genetics. Each of these incremental publications increased size and ethnicity, decreased minor allele frequencies of study populations from previous one, and also addressed the replicability issue in epidemiology. 

 In comparison, our physiological and functional studies of blood pressure QTLs do not need to increase the number of study subjects to be scientifically valid and to achieve sufficient power. This is because our hypertensive rat model and congenic knock in strains derived from it are inbred and homogeneous. They have been studied under a uniform environment. However, we do need to address the reproducibility issue. We have done so by analyzing different QTLs corresponding to different GWAS genes. As a result, our understanding of mechanisms of hypertension pathogenesis may have been improved with newly identified gene candidates responsible for different QTLs.

 While we understand your comments, we also like to bring your attention to our logistics and budgetary restraint of doing research in a 1-investigator-plus-one assistant lab. Just to let you know, we did not divide our results into separate publications, but rather we conducted and reported our progressive search from different stages and in an evolutionary time table. Based on our current work, we are planning further research, once again, in a small scale, in the future. We’ll present our future results as they come. We’d appreciate your evaluation when the time comes.

 We have revised in the introduction to reflect my explanations to you above. ‘Here, we aimed at expanding the scope of rat QTL coverage to capture additional human GWAS genes in progressive stages, since we are a single investigator-driven lab. In the process, our new results have validated the reproducibility of our previous findings. We focused on distinct regions on DSS rat Chromosomes 7 and 8 that contain previously-unexplored rat blood pressure QTLs and human GWAS gene orthologs.

Reviewer #4 comment (2). Line 102 "They are basically the same as documented previously [15]." First, I believe the authors meant to cite reference 14 (not 15). Please verify. Second, this sentence would benefit from being rewritten. Stating that the methods were "basically the same" leaves one wondering if something was different and if so what? Please revise to provide clarity on how the methods differed, and if they did not differ, indicate that the methods were the same.

 Response to Reviewer #4 comment (2): Reference 15 is our original ‘original’ work on QTL modularity. Reference 14 used the same method as reference 15. Now we have added both references to avoid confusion. We have deleted ‘basically’ in the sentence to state that our current method is the same as reported in our previous work.

Reviewer # 4 comment (3). The Discussion section is overly lengthly and has overlap with the original paper (citation 14). Particularly the material on noncoding GWAS SNPs serving as markers for nearby QTLs, a point that was discussed already at length in citation #14. In this manuscript, I very much appreciated the thoughtful Discussion of missing heritability in GWAS and the omnigenic hypothesis. The Discussion needs to be condensed and so reducing content that is similar to the previous publication would shorten the section.

 Response to Reviewer #4 comment (3): You are right. We have deleted the similar discussion as we presented previously that you mentioned. We hope that the revised discussion is acceptable to you.

---

## [Decision Letter · Decision Letter 2]

26 May 2020

PONE-D-19-35461R2

Conserved mammalian modularity of quantitative trait loci revealed human functional orthologs in blood pressure control

PLOS ONE

Dear Dr. Deng,

Thank you for submitting your manuscript to PLOS ONE. After careful consideration, we feel that it has merit but does not fully meet PLOS ONE’s publication criteria as it currently stands. Therefore, we invite you to submit a revised version of the manuscript that addresses the points raised during the review process.

The Reviewers finally agree that the manuscript deserves publication. I recommend that the remaining concerns pointed out by 2 of the 4 Referees are accounted for in view of the definitive acceptance of the paper.

We look forward to receiving your revised manuscript.

Kind regards,

Alessandro Borri

Academic Editor

PLOS ONE

Reviewers' comments:

Reviewer's Responses to Questions

**Comments to the Author**

1. If the authors have adequately addressed your comments raised in a previous round of review and you feel that this manuscript is now acceptable for publication, you may indicate that here to bypass the “Comments to the Author” section, enter your conflict of interest statement in the “Confidential to Editor” section, and submit your "Accept" recommendation.

Reviewer #1: (No Response)

Reviewer #2: All comments have been addressed

Reviewer #3: All comments have been addressed

Reviewer #4: (No Response)

2. Is the manuscript technically sound, and do the data support the conclusions?

Reviewer #1: Yes

Reviewer #2: Yes

Reviewer #3: Yes

Reviewer #4: Yes

3. Has the statistical analysis been performed appropriately and rigorously? 

Reviewer #1: I Don't Know

Reviewer #2: N/A

Reviewer #3: Yes

Reviewer #4: Yes

4. Have the authors made all data underlying the findings in their manuscript fully available?

Reviewer #1: Yes

Reviewer #2: Yes

Reviewer #3: Yes

Reviewer #4: Yes

5. Is the manuscript presented in an intelligible fashion and written in standard English?

Reviewer #1: Yes

Reviewer #2: Yes

Reviewer #3: Yes

Reviewer #4: Yes

6. Review Comments to the Author

Reviewer #1: TO THE AUTHORS:

PLOS One; PONE-D-19-35461.R2

Title: Conserved mammalian modularity of quantitative trait loci revealed human functional orthologs in blood pressure control

In response to my primary concern, the added information provided in the authors’ responses and the statement in the Introduction (Lines 83-84; “We focused…Chromosomes 7 and 8…GWAS gene orthologs”) may be sufficient for future readers.

As a note, the authors’ responses could have been much more direct while simply stating what the new objective(s) is/are and that there is no significant overlap with previous efforts. The reviewer/reader shouldn’t be left “wondering”, “seeming to think”, “hoping for” or “suspecting” anything when trying to understand a scientific study/manuscript. To the reasonable extent possible, a lucid presentation of a body of work as a scientific manuscript is an absolute requirement for publication per the responsibility of the authors. Off the record, limited laboratory staff/funding is indeed a frustrating experience and I wish the best of success for the authors in moving forward with their research. See below for a final suggestion for revising the manuscript.

(1) Introduction, Line 82: Remove “…since we are a single investigator-driven lab” from the sentence as that information is not germane to the objectives and findings of the study/manuscript.

Reviewer #2: Thank you for addressing the comments.i don't have any comment further. The submission looks acceptable.

Reviewer #3: The authors of the manuscript entitled conserved mammalian modularity of quantitative trait loci revealed human functional orthologs in blood pressure control have addressed my comments. The figures are better. I again congratulate Deng & Menard on their revision, and scientific efforts pertaining to the genetics of HTN. In this revision. Their data is complex yet significant, and adds to their existing contribution to the literature.

Reviewer #4: Thank you for the clarification regarding the approach to your investigative program. The revision to the Introduction now clearly distinguishes the purpose of the present manuscript from the previous related work, and the revision to the Discussion reduces the overlap with the previous publication (ref #14). I have only two minor requests remaining:

1. Line 373: The word "topics" should be singular, revise to "topic".

2. Somewhere in the text I would appreciate you noting that the rat has 21 chromosomes. This will help readers understand how much of the rat genome your work has covered and what is remaining.

7. PLOS authors have the option to publish the peer review history of their article (what does this mean?). If published, this will include your full peer review and any attached files.

Reviewer #1: No

Reviewer #2: No

Reviewer #3: No

Reviewer #4: Yes: Daiva Nielsen

---

## [Author Response · Author response to Decision Letter 2]

1 Jun 2020

We have responded to all reviewers`recommendations. We have no changes to make. thanks, Alan

---

## [Decision Letter · Decision Letter 3]

23 Jun 2020

Conserved mammalian modularity of quantitative trait loci revealed human functional orthologs in blood pressure control

PONE-D-19-35461R3

Dear Dr. Deng,

We’re pleased to inform you that your manuscript has been judged scientifically suitable for publication and will be formally accepted for publication once it meets all outstanding technical requirements.

Kind regards,

Alessandro Borri

Academic Editor

PLOS ONE

Reviewers' comments:

Reviewer's Responses to Questions

**Comments to the Author**

1. If the authors have adequately addressed your comments raised in a previous round of review and you feel that this manuscript is now acceptable for publication, you may indicate that here to bypass the “Comments to the Author” section, enter your conflict of interest statement in the “Confidential to Editor” section, and submit your "Accept" recommendation.

Reviewer #1: All comments have been addressed

Reviewer #4: All comments have been addressed

2. Is the manuscript technically sound, and do the data support the conclusions?

Reviewer #1: Yes

Reviewer #4: Yes

3. Has the statistical analysis been performed appropriately and rigorously? 

Reviewer #1: Yes

Reviewer #4: Yes

4. Have the authors made all data underlying the findings in their manuscript fully available?

Reviewer #1: Yes

Reviewer #4: Yes

5. Is the manuscript presented in an intelligible fashion and written in standard English?

Reviewer #1: Yes

Reviewer #4: Yes

6. Review Comments to the Author

Reviewer #1: (No Response)

Reviewer #4: The authors have addressed all of the final comments from the previous review. I have no further requests for revisions.

7. PLOS authors have the option to publish the peer review history of their article (what does this mean?). If published, this will include your full peer review and any attached files.

Reviewer #1: No

Reviewer #4: Yes: Daiva Nielsen

---

## [Editor Report · Acceptance letter]

8 Jul 2020

PONE-D-19-35461R3 

Conserved mammalian modularity of quantitative trait loci revealed human functional orthologs in blood pressure control 

Dear Dr. Deng:

I'm pleased to inform you that your manuscript has been deemed suitable for publication in PLOS ONE. Congratulations! Your manuscript is now with our production department. 

Kind regards, 

on behalf of

Dr. Alessandro Borri 

Academic Editor

PLOS ONE